# Understanding Suicide Risk in People with Dementia and Family Caregivers in South Korea: A Systematic Review

**DOI:** 10.3390/bs12040097

**Published:** 2022-04-06

**Authors:** Jung Won Kong, Ji Young Park

**Affiliations:** 1Research Institute of Social Welfare, Sungkyunkwan University, 25-2, Sungkyunkwan-ro, Jongno-gu, Seoul 03063, Korea; fraukong@skku.edu; 2Department of Social Welfare, Open Cyber University of Korea, C-9F, 353, Mangu-ro, Jungnaug-gu, Seoul 02087, Korea

**Keywords:** people with dementia, family caregivers, suicide risk, systematic review

## Abstract

Dementia-related suicide is not well known. This study aimed to understand the characteristics of suicide risk among people with dementia and dementia family caregivers in South Korea. According to a systematic review of PRISMA guidelines, six electronic databases were searched for research published from 2010 to 2021. Ten studies were included. Among the included studies on people with dementia, one study (25%) showed no increase in the death risk of suicide, while three studies (75%) revealed suicide risk. Furthermore, in the dementia family caregivers, one study (17%) reported suicides and five studies (83%) reported suicide ideation (SI). Early dementia and independence or partial dependence in activities of daily living and home-based care are related to suicide risk in people with dementia. Meanwhile, suicide risk in dementia family caregivers is related to care burden, dementia severity, and poor approaches to coping with the family member’s dementia. The studies reviewed, which demonstrate heterogenous methodologies, suggest that both people with dementia and dementia family caregivers face suicide risk. The results of the current study offer insights useful for the prevention and treatment of mental health issues in both groups.

## 1. Introduction

Some studies reported that, among dementias, Alzheimer’s disease was not associated with a major risk of acting out, except in subjects with relatively concomitant depressive symptoms and evidence of hopelessness [1]. People with dementia are unlikely to attempt suicide because their cognitive impairment impacts suicidal ideation (SI) and implementation; consequently, dementia is not considered a risk factor for suicide [2]. By contrast, the risk appears more clearly in behavioral variant frontotemporal dementia (bvFTD) [3]. SI and suicides have been continually reported among people with dementia in Western and Asian countries [2,4,5,6].

Suicide risk in dementia patients may arise from various factors, such as dementia stage; race; a history of diagnosed depression or other psychiatric illnesses; and previous suicide attempts [4,7]. Notably, the number of suicides among people with dementia in South Korea increased from 555 in 2013 to 598 in 2017; cumulative suicides among people with dementia over this five-year period have been estimated at 2958 which accounted for 3.6% of all suicides among Korean individuals with mental illnesses [8]. 

A notable factor in reducing suicide risk in older adults is the presence of family members who can act as “gatekeepers” [9]. However, dementia family caregivers themselves experience SI [10,11] and homicidal ideation [10,12]. Their care difficulties are associated with uncertain prognosis, treatment, dementia symptom-related stress, and endless care [10,13,14,15]. The health and economic burden of people with dementia under the age of 64 years is rapidly increasing in the city of Seoul, South Korea [16]. The burden of caring for a family member with dementia increases as the symptoms worsen in people with dementia [14]. Support costs for a patient with very mild dementia have been reported to be twice as low as for a patient with severe dementia [17]; the more severe the dementia, the greater the financial burden on the family. The care burden is highly likely to be linked to the primary caregivers’ depression [14]. In a previous study [18], 75% of family caring for people with dementia reported depression.

Moreover, although depression and anxiety management are required for suicide prevention in people with dementia [4], there is lack of a professional medical system that can treat and manage the mental health of people with dementia [15]. In addition, it has been reported that people with dementia and dementia family caregivers differently exhibit SI [2,4,10,11]. It is known that dementia family caregivers have high caring stress, but the factors that put them at risk of suicide have rarely been studied.

Despite people with dementia and dementia family caregivers facing problems that may lead to suicide, research on suicide risk in these groups is lacking in both Western and Korean contexts. Although some studies have examined suicide risk among people with dementia and dementia family caregivers in Korea, to the best of our knowledge, no systematic reviews have been conducted. Korea is a rapidly aging society with a growing number of people with dementia; therefore, it is crucial to develop a comprehensive understanding of suicide risk in people with dementia and dementia family caregivers in Korea. Thus, this study aimed to synthesize the characteristics of suicide risk for people with dementia and dementia family caregivers in Korea over the last ten years (2010–2021).

## 2. Materials and Methods

### 2.1. Literature Search Strategy

This study complied with the Preferred Reporting Items for Systematic Reviews and Meta-Analyses (PRISMA) checklist [19]. Six search engine databases—the Korea Citation Index (KCI), earticle, Google Scholar, Web of Science, PubMed, and PsycINFO—were used for this review. A search was performed with (AND/OR/NOT). KCI, earticle, and Google Scholar were searched for Korean-language studies using Korean keywords, including (dementia AND suicide) OR (dementia elderly AND suicide) and (dementia family AND suicide) OR (dementia family AND suicide risk). The Web of Science, PubMed, and PsycINFO databases were searched using English keywords, including (dementia AND suicide AND South Korea) OR (dementia elderly AND suicide AND South Korea) and (dementia family AND suicide AND South Korea) OR (dementia family AND suicide risk AND South Korea).

### 2.2. Eligibility Criteria

The authors set the inclusion and exclusion criteria for the selected studies. The inclusion criteria were as follows: (1) target of people with dementia and dementia family caregivers in South Korea, (2) suicide risk outcome (e.g., suicidal behavior or SI) in the target population, (3) qualitative and quantitative methods (e.g., suicide-related case analyses, cross-sectional studies, cohort studies), (4) articles with original data (i.e., not reviews, systematic reviews, or meta-analyses), (5) peer-reviewed published studies written in Korean or English, and (6) published between 2010 and 2021. The exclusion criteria were as follows: (1) did not present suicide-related outcomes of people with dementia and dementia family caregivers, (2) non-South Korean participants, and (3) published outside the study extraction period. 

### 2.3. Analytical Process

The study selection process was recorded using an Excel spreadsheet and is illustrated using the PRISMA flow chart [20]. Studies were selected for review only if the authors reached a consensus. Collected data included information on SI, SA, and suicides among people with dementia and dementia family caregivers and the included studies collected data using both qualitative and quantitative methods. 

### 2.4. Study Selection

Six search engines were used to search for the above mentioned keywords. Study selection was performed by each author and duplicate studies were removed. Thereafter, the titles and abstracts were evaluated based on the inclusion and exclusion criteria, such as measurable outcomes of suicide risk and the participation of people with dementia and dementia family caregivers.

### 2.5. Quality Appraisal and Assessment

The types of studies in the review included cohort, qualitative, and cross-sectional studies. Thus, the present study used the Mixed Methods Appraisal Tool (MMAT), which is used to assess the quality of qualitative, quantitative, and mixed research [21]. The MMAT includes criteria such as whether the research questions and data are presented clearly and in a suitable manner for the type of research. Both authors performed quality appraisals on the selected studies using the MMAT. When the authors disagreed, a final decision was made after discussion. 

### 2.6. Data Extraction and Synthesis

The authors performed data extraction on the included studies based on the Cochrane data extraction forms [22]. The data extraction process was performed according to the study focus, study type, characteristics of the dementia family caregivers and people with dementia, prevalence of suicide risk, risk or protective factors, and outcomes. Each author conducted data extraction for all included studies and recorded the results in an additional file. Subsequently, the differences between the authors’ data extraction results were analyzed and resolved through reconfirmation and communication.

## 3. Results 

### 3.1. Search Outcome

The initial search yielded 143 studies with the required keywords; 48 studies remained after removing duplicates. Assessment of the selected studies based on whether they met the inclusion and exclusion criteria for this review resulted in 11 studies remaining. During the study selection process, one study involved suicide prevention and was thus excluded as it did not meet the criteria. Finally, 10 studies with full text were selected for this review (Figure 1). 

### 3.2. Quality of Methodology

The Appendix A presents the quality assessment results for each included study (Table A1). In this review, the criteria for qualitative studies and quantitative non-randomized studies were included among the MMAT methodological quality criteria according to the study design. The MMAT quality appraisal was applied to qualitative studies (*n* = 2), including news article analyses, cohort studies (*n* = 3), and quantitative studies (*n* = 5). Most studies were evaluated as appropriate.

### 3.3. Characteristics of Included Studies on People with Dementia and Dementia Family Caregivers

Of the ten studies in this review (Table 1), seven were written in Korean [6,12,23,24,25,26,27] and three in English [5,28,29]. All included studies analyzed Korean data. Four studies focused on people with dementia [5,6,28,29], analyzing the characteristics of SI [6] and suicides [5,28,29] in people with dementia using quantitative and cohort research methods. Six studies focused on dementia family caregivers [12,23,24,25,26,27]; of these, four investigated SI using quantitative methods [24,25,26,27] and one used focus group interviews and addressed dementia family caregivers’ SI [12]. Only one study conducted a content analysis of newspaper articles to uncover suicides in dementia family caregivers [23]. Of the six dementia family caregiver studies, one study focused only on spouses [26] and one [24] on adult children; the remaining studies focused on both spouses and adult children [12,23].

### 3.4. People with Dementia

#### 3.4.1. Gender and Age

Kim and Hyun [6] studied 298 participants who were long-term care registered in nursing and home-based care facilities with other condition. One study [5], which used the National Health Insurance Service Senior Cohort data from 2004 to 2012 for people with dementia, included 36,541 people with dementia. In both studies [5,6], women comprised a higher proportion of the sample than men. Regarding age, the mean age was 84.1 years in one study [6] and people with dementia aged 75 years or more accounted for 69.5% of the sample in the other study [5]. Two other studies [28,29] addressed the gender and age composition of suicide in people with dementia or mild cognitive impairment. Men accounted for 42.2% in one study [28], with a mean age of 71.42 years. In the other study [29], the suicide rate for men was higher than that for women, and the suicide rate for those in their 60s was higher than that for those in their 80s. 

#### 3.4.2. Dementia Severity and Heath Condition

In four studies, participants were restricted to people diagnosed with dementia, prescribed antidementia drugs, registered with the Clinical Research Center for Dementia of Korea, or with a dementia related conditions [5,6,28,29]. In three studies, Mini-Mental State Examination (MMSE) scores were used as a criterion for participation or suggested [5,6,28]. One study indicated an MMSE-KC score was between 15 and 23 [6]. The other study used an MMSE score ≤26 [5]. Regarding suicides among people with dementia, the Korean Mini-Mental State Examination (K-MMSE) mean SD score was 21.29 [28].

In South Korea, the long-term care grade can identify the level of support required for people with dementia such as performing activities of daily living (ADL) and instrumental activities of daily living (IADL). In two studies [6,29], the long-term care service use or eligibility were presented. 

#### 3.4.3. Risk Factors Influencing suicides and SI in People with Dementia

Although both studies [5,6] dealt with the relationship between mental health and suicide risk, different results were reported. A positive association between depression and SI was determined in the study by Kim and Hyun [6]. However, the other study found that suicide was associated with dementia rather than mental health problems [5]. The associated risk factors of SI for people with dementia were depression, previous SA, at-home care, and partial ADL dependency [6]. In the other study, time since dementia diagnosis was associated with suicide risk [5]. Meanwhile, in both studies, suicide risk was found for those who received care at home than for those who received care in an institution [6,29]. Moreover, conditions that did not result in full ADL dependence were identified with suicide risk in two studies [6,29].

#### 3.4.4. Prevalence of Suicides and SI in People with Dementia

One study reported a mean SI score of 5.7 (range: 0–20) and an SA history of 9.9% [6]. Another study revealed that the highest number of suicides (46 suicides) occurred within one year of dementia diagnosis and decreased to 29 during the second year [5]. Furthermore, suicide risk increased more in non-users of long-term care services compared to users [29]. Of the included studies, only one found no increased death risk related to suicide in people with mild cognitive impairment or people with dementia [28].

### 3.5. Dementia Family Caregivers

#### 3.5.1. Characteristics of Dementia Family Caregivers (Gender, Age, and Dementia Family Caregivers Composition)

In one study, with regard to the types of suicide, completed suicides, completed suicides after homicide, and companion suicides were reported [23]. Across these types, the suicide rate was higher among males, especially for suicide after homicide [23]. In four studies on SI [12,24,26,27], the number of female participants was higher than the number of male participants.

Regarding age, suicide after homicide was reported in 80% of individuals aged 60 years or older, companion suicide in 71.4%, and completed suicides in 55.6% [23]. In one study, three dementia family caregivers were over 60 years old and three were between 42 and 59 years old [12]. There was a difference in the age of dementia family caregivers according to family composition among the included studies. The average age of spouses in one related study was 75.58 [26] and 68.7% of adult children and daughter-in-law were aged 46 to 55 years in another related study [24].

Family member participation composition across the studies was as follows: In a case-based study, four adult children, one daughter-in-law, and one spouse participated [12]. Meanwhile, population-based studies ([24,26,27]) involved 160 spouses [26] and 326 adult children and daughter-in-law [24], and 71.3% of spouses lived with people with dementia [27]. 

#### 3.5.2. Long-Term Care Grade and Care Duration

Three studies presented the composition of long-term care grades for people with dementia [12,24,26]. Across these three studies, there were more people with dementia with long-term care grades of three and four than with a long-term care grade of five or no grade [12,24,26]. Two studies reported the length of care and the illness period [24,26]. The duration of care was, on average (Mean), 11.48 h a day [26], and the duration of illness was reported in 78.9% as one year to four years [24].

#### 3.5.3. Risk and Protective Factors Influencing Suicide Risk for Dementia Family Caregivers

Suicides were associated with the care burden of dementia family caregivers, the burden of other family members, feeling hopeless regarding recovery, violent behavior and language problems of people with dementia, and absence of alternative caregivers [23]. The risk factors for SI among dementia family caregivers included the absence of alternative caregivers and care burdens (e.g., endless caring and economic, physical, and mental burdens) [12,24,25]. The focus group interview study reported impulsive SI due to their care burdens; dementia family caregivers expressed a sense that the care burden would only end if they died by suicide [12]. Suicide risks related to dementia symptom level and severity were also found [25,26]. In one study on spouses, feelings of entrapment and poor coping strategies were associated with SI [26]. The combination of “high risk-low protective factors” involved the highest levels of SI [27].

Only two studies presented protective factors related to SI among dementia family caregivers. One study detailed that such factors included social activities and resource as protective factors of SI among family living with people with dementia [27]. Another study presented self-efficacy as a protective factor in the relationship between care stress and SI among adult children and daughter-in-law serving as dementia family caregivers [24]. 

#### 3.5.4. Prevalence of SI, SA, Suicides among Dementia Family Caregivers

Suicidal behavior among dementia family caregivers occurs in diverse forms [23]. In a study consisting only of spouses, although the methodology and sample size were heterogeneous, the reporting rate of SI was higher than that of other studies [26]. 

## 4. Discussion

This review aimed to explore suicide risk characteristics among people with dementia and dementia family caregivers in South Korea over the last ten years (2010–2021). Ten studies were included; within this review, there were fewer studies on people with dementia than on dementia family caregivers. Suicide risk was reported for both people with dementia and dementia family caregivers, but the characteristics of suicide risk were different, a finding that is consistent with the results of a previous study [2,4,10,11]. In terms of the related suicide risk among people with dementia, there were 21 published studies spanning 35 years (1980–2015) in a previous systematic review of Alzheimer’s disease, compared to four published studies for people with dementia across 10 years in the current study [30]. Furthermore, the previous review included eight cross-sectional and two longitudinal studies [30], whereas in the current review, only one of the four studies was cross-sectional; the other three were cohort studies. While a prior meta-study found that the effect size of SA was higher than SI among people with dementia [31], the current review found that SA was rarely assessed for people with dementia.

Second, regarding the demographics of dementia family caregiver participants, there generally tended to be more female participants than male participants. However, a study on completed suicides in dementia family caregivers reported a higher prevalence in men than in women. This review revealed gender-related differences between study participants and actual completed suicides among the dementia family caregiver studies. In terms of the social context, since Korean elderly males have in the past been considered breadwinners, they may have lacked care resources for people with dementia as compared to women. Regarding the age of people with dementia, it was difficult to estimate the average owing to the heterogeneity of methods across studies. The dementia family caregiver studies revealed age differences according to family member composition. 

Third, prevalence, risk factors, and protective factors for completed suicides, SI, and SA varied due to methodological heterogeneity among the studies. A previous cohort study on SI in people with dementia in eight European countries revealed nationality and depression as suicide risk factors [32]. However, this review showed different results for the associated mental health risks [5]. In addition, the preceding study [32] reported that SI among people with dementia decreased over time and two studies [5,28] in this review were considered in a similar context. The review findings were considered in light of previous studies of suicide risk factors associated with people with dementia such as depression, experience with suicide, and early dementia stage [4,7,32].

Similar to the findings of previous studies [10,11], dementia family caregivers were also exposed to suicide risk. In addition, the factors of suicide risk for dementia family caregivers were similar to those reported in previous studies [10,13,14].

Based on the results of this study, the following recommendations are suggested: Dementia is seen as damage to the nervous tissue in the brain [33] and includes not only memory impairment but also emotional problems [14]. Mental health for people with dementia requires treatment by specialists [14,15]. However, very few hospitals offer specialized mental health care for people with dementia in South Korea [34]. Mental health services for suicide prevention for early-diagnosed people with dementia who are relatively cognitively and physically healthy should be considered. In this review, the scale for SI measurement was identified in four studies [6,24,25,26] and the various measuring SI tools were not used in the included studies. At the various levels, depression, SI, and suicide risk assessments for suicide prevention in dementia patients and caregivers should be considered [35,36,37]. It is also necessary to consider a combined treatment approach, such as robotic and cognitive therapy, to reduce the risk of suicide in people with dementia [38].

Seong et al. [34] reported a lack of employment support for dementia family caregivers. Our review found that the employment status of adult children of people with dementia was related to SI [24]. Support is needed so that dementia family caregivers can balance work and care. However, one study identified bondage from dementia spouse care as a suicide risk. In South Korea, 2.5 times more women had dementia than men in 2019 [39]. More recently, spouses are more likely to become dementia family caregivers than adult children and support for spouses as dementia family caregivers is essential. Based on the results of this review, specific support to alleviate the care burden (e.g., financial support, diversity of care resources, and educational opportunities for coping with the behavioral changes of people with dementia) needs to be introduced.

This study involved several limitations. Regarding suicide risk among people with dementia and dementia family caregivers, the heterogeneity of methodologies prevented us from synthesizing robust new knowledge. Additionally, although the researchers carefully extracted the literature, the possibility of omission cannot be excluded. As the included studies are peer-reviewed studies, biased results may have appeared due to publication bias. Nevertheless, the major contribution of this review is that it addresses the lack of knowledge about suicide risk in people with dementia and dementia family caregivers.

## 5. Conclusions

People with dementia and dementia family caregivers experienced suicide risk after a dementia diagnosis or as the severity of dementia worsened. The findings of our review expand existing knowledge on dementia-related suicide risk literature by offering insights on dementia-related suicide risk among Koreans. 

Notably, we found that there were fewer studies on SI than on suicides among people with dementia. Accordingly, it was difficult to obtain sufficient knowledge about suicide risk in people with dementia. The limitations of conducting research on suicide risk in people with dementia may include the cognitive status of people with dementia or difficulty in recruiting research participants. Consequently, additional research must be conducted to monitor and manage suicide risk among people with dementia. Moreover, few studies in this review measured SA. Additional assessment of SA among people with dementia and dementia family caregivers needs to be conducted. Moreover, it is necessary to expand the data for future research by using data from Korea and other countries.

## Figures and Tables

**Figure 1 behavsci-12-00097-f001:**
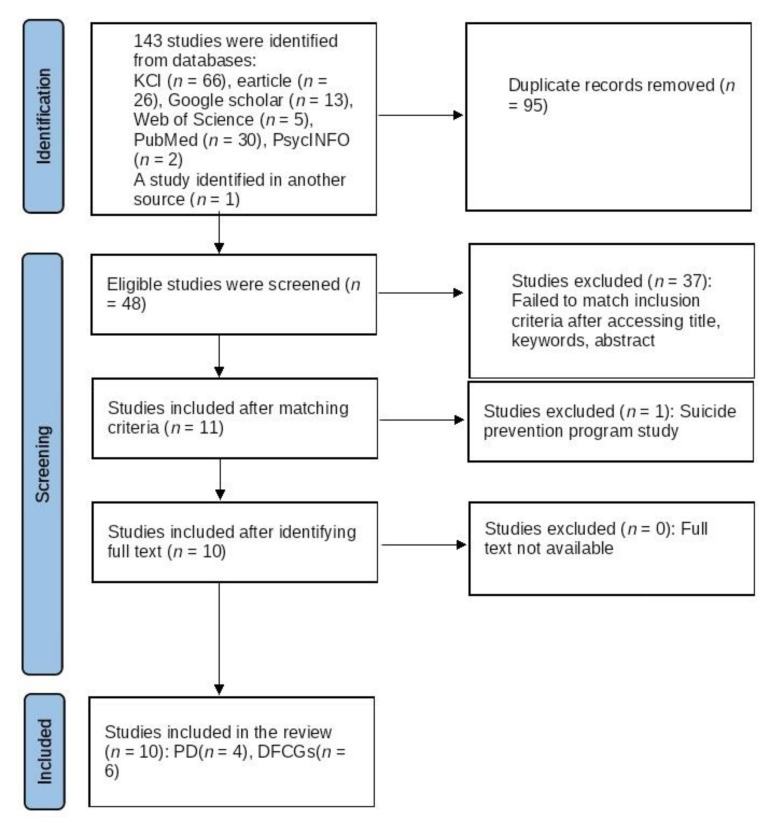
PRISMA flow diagram of studies on suicide risk among people with dementia and dementia family caregivers. PD: People with dementia. DFCGs: Dementia family caregivers.

**Table 1 behavsci-12-00097-t001:** Characteristics of suicide risk among people with dementia and dementia family caregivers.

Author (Year), Country	Focus	Study Type	Participant Characteristics	Suicide Risk Characteristics				
				SI/SA (No, %)	Completed Suicides (No, %)	Risk Factors	Protective Factors	Outcome
Kim & Hyun (2013) [6], South Korea	SI (people with dementia)	Stepwise multiple regression (quantitative study)	298 people with dementia; Female (80.8%) Male (19.2%); Age mean value: 84.1	SI mean score: 5.70 SA: 9.9% Suicidal Ideation Scale		Depression, history of SA, home care, partial support required for ADL		Positive association between depression and SI
An, Lee, Jeon, Son, Kim, & Hong (2019) [28], South Korea	Death risk of suicides in people with dementia or mild cognitive impairment	SMR, Cox regression (cohort study)	10,169 patients from the Clinical Research Center for Dementia of Korea (2005–2013)		Suicides: 0.44% *(n* = 45) Among suicide death, 42.2% male and mean age of 71.42	Unemployment	Increased in educational period	No increase in risk of death caused by suicide
Moon, Choi, & Sohn (2021) [29], South Korea	Suicide (people with dementia)	Kaplan–Meier and Cox regression (cohort study)	62,282 elderly people from Older Adults Cohort DB (2002–2015)		25.94 per 100,000 people in suicide mortality rate of annual average	Significant differences in suicide risk across conditions		Long-term care service user suicide risk lower than that of a non-user in the prior to the expansion of the dementia grade
Choi, Lee & Han (2021) [5], South Korea	Suicides (people with dementia)	A time-dependent Cox proportional hazards model (cohort study)	Including 36,541 people with dementia from the National Health Insurance Service Senior Cohort (from 2004 to 2012); Among people with dementia, male (30%) and female (70.0%); Age of 75 years and above 69.5%		First year after dementia diagnosis showed the highest suicide rate of 125.9 per 100,000 people	Dementia diagnosis		Higher in suicide risk in dementia group compared to group without dementia within one year after of the diagnosis
Kim, (2014) [23], South Korea	Suicides (dementia family caregivers)	Newspaper article analysis (case analysis)	Including 26 suicide-related cases in news articles from 1920 to 2014. The family composition included spouse, son, daughter, and daughter-in-law etc.		Identifying nine suicides, ten suicides after homicide by family, and seven companion suicides with people with dementia	Care burden, absence of alternative caregivers and loss of hope of getting better for dementia		Suicide after homicide demonstrated the highest suicides of the three suicide types
Kim & Um (2015) [12], South Korea	SI (dementia family caregivers)	Focus group interview (qualitative study)	Six dementia family caregivers Care for mother (4), spouse (1), and mother-in-law (1); five female and one male; three aged 42–59 years and three aged 65–75	SI		SI related to care stress and burden		Impulsive SI due to burden of care
Kim, Kim, Jang & Song (2016) [25], South Korea	SI (dementia family caregivers)	Mediation effect analysis (quantitative study)	415 dementia family caregivers	SI: 21% SA: 6.7% Suicide Ideation Scale		Dementia symptom level		Significant partial mediating effect of care burden on the relationship between dementia symptom level and SI
Park (2018) [27], South Korea	SI (family living together)	Latent hierarchical model analysis Binary logistic analysis (quantitative study)	2715 dementia family caregivers; Male (45.5%) and female (54.5%); Average age: 60.08 Spouse: 71.3%	SI: 17.1% (*n* = 465) SA: *n* = 15		Three sectors of risk factor in economic, physical, and mental domains	Four sectors of protective factors in social related factors	Combination of “high risk-low protection factors” showed the highest SI
Jeong (2017) [24], South Korea	SI (adult children and daughter-in-law)	Multiple regression analysis (quantitative study)	326 adult children and daughter-in-law; Male (30.7%) and female (69.3%); Age 51–55: 35%	SI: 32.6% Suicide Ideation Scale		Positive association between care stress and SI	Self-efficacy	Partial mediating effect of self-efficacy in the relationship between stress and SI on dementia family caregivers
Du & Han (2018) [26], South Korea	SI (spouse)	Multiple regression analysis (quantitative study)	160 spouses; Male spouse (35.6%) and female spouse (64.4%); Average age: 75.58	SI: 45.5% Suicide Ideation Scale		Higher entrapment, poor dementia severity, lower coping strategies		Influencing factors of SI among spouses of dementia family caregivers

Note: ADL: activities of daily living; SI: suicidal ideation; SA: suicide attempt; SMR: standardized mortality ratio.

## Data Availability

The data is available upon reasonable request.

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
