# Peer review of "Understanding Suicide Risk in People with Dementia and Family Caregivers in South Korea: A Systematic Review"

_behavsci, 2022, doi:10.3390/bs12040097_

Round 1
Reviewer 1 Report
The presented article reflects current issues and deals with the understanding of suicidal tendencies in individuals suffering from dementia.
First of all, I would like to focus on a more detailed description in the case of Chapter 1 (Introduction). I believe that in this case, the theoretical basis should be expanded by more sources in the field of psychiatry and substantiate the information with statistically significant information, ie that the number of suicides or attempts in individuals suffering from dementia is increasing. There is also a need to better define what is the cause? Poor quality health services? Separation of the individual from the family? Social isolation? The answer to these questions would enrich the article.
I have no major reservations when presenting the results.
The authors presented a systematic review in a very professional quality. The presented information corresponds to the intention of the article. I consider this part to be very high quality and beneficial. It is a pity that these are localized data and the issue has not been examined from a broader perspective (but I understand the difficulty of obtaining data). The authors also suitably supplemented the research survey with the Prisma flow diagram, which offers better orientation in the individual steps of the research survey.
In the discussion, I would like to recommend authors to cite sources that would underline the importance of combination therapy (eg this article can be used and formulated in general: https://www.nel.edu/the-effect-of-combined-therapy-on-the-support-and-development-of-social-skills-of-people-with-multiple-sclerosis-in-senior-age-2753/ and https://www.nel.edu/combined-therapy-for-patients-after-ischemic-stroke-as-a-support-of-social-adaptability-2714/).
In Europe, the combination of different therapeutic approaches is the trend towards reducing the risk of suicidal behavior, so I recommend adding.
After the addition, I have no further reservations, the article is beneficial.
Author Response
"Please see the attachment."

Reviewer 2 Report
The manuscript of Kong et al. provided an in-depth review of suicide risks of dementia patients and their caregivers in modern Korean society. Such investigations, whilst common in Western psychiatric and geriatric studies, were noticeably less affluent in the context of Far East societies as the result of strong cultural differences. Hence, this article provided a missing piece for anyone who would like to explore the connection between neuropsychiatric disorders, public healthcare, and the associated suicide risk of the patients and their caregivers in Korean society. However, some minor language issues require clarification before accepting the manuscript for the publication process, as noted below.
Line 34: "may be due" -> may arise from;
Line 45-46. This sentence is self-contradictory. How could "mild dementia...to be twice as high as...severe dementia" whilst "the more severe...the more financial burden" both proven as true? Please revise;
Line 199: "In one qualitative study..." I do not see the underlying argument of this sentence. Please revise;
Line 214-217: Case-based studies([12]) and population-based studies([24, 26, 27]) should be separately discussed in two sentences. The current form lead to significant confusion;
Line 263-264: "Furthermore..." I could not get the point of this sentence. Please elaborate.
Author Response
"Please see the attachment."

Reviewer 3 Report
Please see the attached file.
Thank you very much.
Best regards.

Author Response
"Please see the attachment."
